# Impacts of Floods on Agriculture-Dependent Livelihoods in Sub-Saharan Africa: An Assessment from Multiple Geo-Ecological Zones

Roland Azibo Balgah [1], Kester Azibo Ngwa [2], Gertrud Rosa Buchenrieder [3,*] and Jude Ndzifon Kimengsi [4,5]

1 College of Technology, The University of Bamenda, Bambili P.O. Box 39, Cameroon
2 Higher Institute of Agriculture and Rural Development, Bamenda University of Science and Technology, Nkwen P.O. Box 277, Cameroon
3 Institute of Sociology and Economics, RISK Research Center, Universität der Bundeswehr München (UniBwM), 85577 Neubiberg, Germany
4 Faculty of Environmental Sciences, Technische Universität Dresden, 01069 Dresden, Germany
5 Department of Geography, The University of Bamenda, Bambili P.O. Box 39, Cameroon
* Correspondence: gertrud.buchenrieder@unibw.de

**Abstract:** Surging extreme events, particularly floods, have stimulated growing research on their epidemiology, management, and effects on livelihoods in Sub-Saharan Africa (SSA), especially for agriculture-dependent households. Unfortunately, the topical literature is still characterized by independent, isolated cases, with limited relevance to understanding common flood effects across geographical space and time. We bridge this knowledge gap by analyzing the effects of multiple cases of flash, coastal and riverine-cum-pluvial ('complex') floods on agriculture-dependent livelihoods in three (Sudano Sahelian, Coastal and Western Highlands) geo-ecological zones in Cameroon. The analysis makes use of a sample of 2134 flood victims (1000 of them in the Sudano-Sahelian, 242 in the Coastal, and 892 in the Western Highlands zones) of 26 independent community floods: 11 in the Sudano-Sahelian, 3 in the Coastal, and 12 in the Western Highlands zone. Irrespective of flood type and geo-ecology, agriculture-dependent livelihoods were gravely impaired. However, the impacts on livelihoods and public goods (such as road or communication systems) significantly varied in the different geo-ecological zones. The study concludes with the need to include context-specificity in the flood impact assessment equation, while identifying common effects, as is the case with agriculture in this study. We emphasize the need to up-scale and comparatively analyze flood effects across space and time to better inform flood management policies across SSA.

**Keywords:** floods; agriculture-dependent livelihoods; impacts; geo-ecological zones; sub-Saharan Africa

## 1. Introduction

Livelihoods around the world are affected by increasing extreme events, which depend on several factors; these include vulnerability, hazard intensity and duration, risk perceptions and exposure [1–11], system resilience and response capacity, weak disaster management institutions [12–14], and ineffective coping mechanisms [1,3,13]. Other influencing reasons include the degree of correlation in the affected group (that is, whether the event is idiosyncratic or covariate), and whether the management strategies are applied *ex-post* (before the event) or *ex-ante* (after the event) [15]. Overall, the effects of extreme events on the environment and on the livelihoods of affected populations are negative [15–20].

The concept of livelihood has gained importance in the last three decades and has been the focus of discourse across disciplines on poverty reduction, development policy, sustainable resource management, and climate change [21]. Defined in a broader sense, a livelihood comprises the capabilities, assets (including both material and social resources) and activities required for living. A livelihood is considered sustainable "when it can cope





with and recover from stresses and shocks, maintain or enhance its capabilities and assets, while not undermining the natural resource base" ([22], p. 175).

Agriculture remains a major source of livelihoods, especially for rural households in developing countries [23,24]. Globally, approximately 2.5 billion people, of which 60% reside in developing countries, depend almost entirely on agriculture for their livelihoods, generating over half of the global food production on small farms [25]. Global agricultural production and agriculture-based livelihoods are particularly at risk, due to rising extreme events, such as floods and droughts [26]. Therefore, understanding the agriculture-dependent livelihoods–extreme events nexus, is relevant to developing resilience, adaptation, mitigation or coping strategies that can permit agrarian societies to survive as extreme events become more frequent [16,25].

The rapid surge of extreme events has stimulated research on their epidemiology and on their multifaceted effects concerning the economy [18], society [7,27], the environment [28,29] and livelihoods [10,15,19]. For instance, in the last seven decades, natural disasters have caused an estimated global economic loss of over USD 3 trillion, have inflicted over 1.3 million casualties, and impaired over 4.4 billion people [16]. Frequently reported effects include the loss of human and animal life, and the loss of livestock, crops, land, houses, and infrastructure. Furthermore, natural disasters force displacement [3,7–9,13,20,30,31], impair health conditions and disrupt the supply of critical services, such as electricity and medication [2,17–20]. These outcomes directly or indirectly interrupt livelihoods in the impacted communities. In 2020, EM-DAT, the international disaster database (see: www.emdat.be/), recorded 389 environmental disasters, which caused the loss of 15,080 human lives, affected 98.4 million people, and inflicted financial losses of over USD 171.3 billion [24].

Floods are one of the most frequent and virulent extreme events worldwide. Their frequency is linked to the consequences of climate change and socio-economic development [7,15,19,26]. For many decades, floods have accounted for most of the global effects of natural disasters on economic growth and livelihood outcomes [7–9]. In fact, flooding was the major source of recorded global disasters between 2000 and 2019; floods were also the second largest natural disaster after droughts, in terms of the total number of affected persons over the same period [7,9].

Floods have occupied the premier rank among global environmental disasters in the past twenty years in terms of the frequency of their occurrence; floods account for 44% of all the registered disaster events [7–9,26], and top the list of natural disasters in terms of economic damages: they cost USD 651 billion in this time-span [7]. Floods were only second to drought, with 1.6 billion people being affected worldwide [7,9].

Seven million people were affected by floods in Africa in 2020; this was the highest impact on record since 2006. The bulk of the effects are recorded in Sub-Saharan African (SSA) countries [32], in which the pervasive flood-related devastation of livelihoods is expected to surge as the frequency of events increases [33,34]; this is in a context characterized by a weak formal and informal institutional capacity for disaster management [28]. This is expected to retard economic and social change, in that it will scale back the attained progress in reducing poverty, and negatively affect the global capacity to achieve the sustainable development goals (SDGs) of the Global Agenda 2030 [35]. These expectations have provided the impetus for a strong flood research agenda in SSA [3,19,36,37].

One way to reduce flood effects in SSA is to develop an overarching policy agenda for flood management. For instance, the Sendai Framework for Disaster Risk Reduction (2015-30), one of the first major agreements of the post-2015 Global Agenda, supports countries in identifying and implementing concrete actions to protect development gains from the risk of various disasters (www.undrr.org/implementing-sendai-framework/what-sendai-framework, accessed on 15 January 2023). Such an overarching policy agenda for flood management will greatly benefit from empirical research and the identification of robust trends for all cases, space and time [11]. Understanding the impacts of flooding, beyond the individual case and the different geo-ecologies, provides valuable insight into effective policy decisions in SSA. In these areas, poverty is endemic and flooding severely

impacts agriculture, which is the mainstay of most people in SSA [7,11]. The flood research landscape in Africa, and particularly in SSA, is flourishing [38]. Unfortunately, research on floods is dominated by isolated case studies [19], which provide limited insights and are of limited relevance for flood risk reduction, mitigation, and adaptation at national, regional and sub-continental levels. Comparative analyses still beg for attention. A shift towards multiple case studies that transcend geo-spatial limits can significantly influence flood management policy decisions at aggregate levels.

In this context, systematic analyses of nationally aggregated data can provide directives for both national and (sub) continent-wide policies; SSA urgently needs these, given her growing subjection to floods [16,32,39,40]. However, very little has been done to compare the effects of multiple floods, and even less across different geo-ecological zones. We narrow this knowledge gap by analyzing the effects of multiple flood types (e.g., flash, coastal and 'complex' or riverine-cum-pluvial) on agrarian livelihoods in Cameroon. The floods studied occurred independently of each other in three geo-ecological zones in Cameroon, that is, the Sudano-Sahelian, the Coastal (Humid Forest with Monomodal Rainfall), and the Western Highlands (Montane or Western High Plateau) zone. The three zones are marked by distinct specificities in terms of the frequency and triggers of flood disasters, and by distinct geographical characteristics. The underlying denominator for all the studied geo-ecological zones is the fact that agriculture is the main source of livelihoods for the rural population [41].

The major scientific contribution of this study is achieved by analyzing the effects of multiple, independent floods on agriculture-dependent livelihoods in three Cameroonian geo-ecological zones in a single study. Our contribution, therefore, provides initial reflections and insights on the effects of extreme events, particularly floods on agriculture-based livelihoods across space and time; it also stimulates reflections on possible (research and policy) perspectives to develop flood-resilient livelihood systems in communities in which agriculture is the mainstay. To achieve this, the study is guided by a central research question: Do the impacts of floods among agriculture-dependent livelihoods differ by geo-ecological zone?

Section 2 presents a concise overview of the impacts of floods on livelihoods. Then, Section 3 depicts the study sites in the three geo-ecological zones, as well as the sampling-cum-data collection approaches. In Section 4, demographic results are presented first, followed by the impacts of floods on livelihoods. Concluding remarks are summarized in Section 5.

## 2. Conceptual Framework for Assessing Flood Impacts on Livelihoods

### 2.1. Brief Outline of the Conceptual Sustainable Livelihood Framework

Since the 1970s, the farming system research approach has been continuously broadened to encompass a wider set of issues, resulting in the Sustainable Livelihoods Framework (SLF). The livelihoods approach offers a rounded, bottom-up perspective and strives for a more holistic, people-centered approach. Through the concepts of 'vulnerability' [42], 'sensitivity', and 'resilience' [43], the SLF also seeks to capture the hazards that (farm) households face, the shocks that these engender, and their capacities to respond to them [44].

The SLF integrates all the important aspects that affect livelihoods. A widely accepted definition stems from Chambers and Conway ([45], p. 7–8): "a livelihood comprises the capabilities, assets and activities required for a means of living". Capabilities refer to the set of alternates that an individual can attain with her/his/their economic, social, and personal characteristics. The SLF emphasizes the asset pentagon (see Figure 1), consisting of natural, physical, human, financial and social assets. Access to these assets, as well as to their efficient use, determine the resilience of the right-holders vis-à-vis extreme events. The asset pentagon is embedded in additional impacting factors, such as the 'vulnerability context' (e.g., demographic trends, depletion of natural resources, extreme events, etc.), 'structures and institutions' (e.g., gender roles, private and public disaster risk management frameworks), and 'intention and behavior' (e.g., agency resulting from perceived risks).

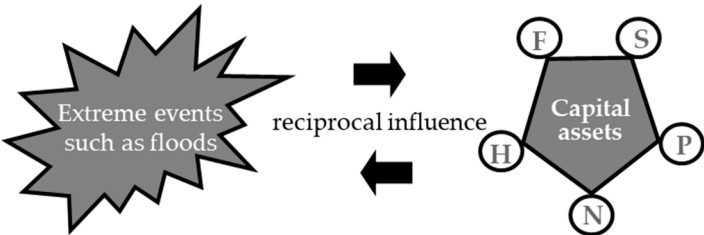

**Figure 1.** Capital asset pentagon of the SLF. Source: Excerpt from Buchenrieder and Möllers ([44], p. 18). Notes: S = Social capital, i.e., social networks. F = Own financial means and access to external finance. N = Natural resources, e.g., land. P = Physical assets. H = Human capital, e.g., labor, health.

*2.2. Brief Literature Review on the Impacts of Flooding on Household Livelihoods*

Irrespective of, and sometimes independent of literary positions and classification approaches [16], floods continue to have serious direct and indirect human, physical, social, economic, psychological and other effects on their victims. The most common direct effects include the destruction of houses, crops, livestock, agricultural land, loss of lives and forced displacement [1–3,8,19,23,24,29,36,46–51]. Indirect effects may include post traumatic/mental disorders, and an increased frequency of diseases. Floods increase vulnerability, especially in developing countries, increasing the exposure of people to other livelihood shocks, such as economic and political crises [19,47,50]. Floods can perpetuate poverty by damaging goods and possessions, by causing clean-up costs [49], and by causing the loss of livelihood resources, for example, as a result of forced migration [28,31,52]. Those who do not migrate due to place-based attachment [31], a lack of capacity [52] or information asymmetry about the possible outcome of forced environmental migration [28], often witness livelihood degradation. Floods, therefore, affect economic/financial capital accumulation directly by destroying productive assets, such as livestock and crops, and indirectly by engendering income loss from not being able to liquidate lost assets [27,53].

Floods can puncture the accumulation of human capital, especially for agriculture-dependent households. For instance, Zerihun and Befikadu [3] reported that floods in North Western Ethiopia affected the human capital of flood victims in the form of reduced health conditions, the destruction of education systems, as well as the loss of skilled labor. Saleh [47], as well as Musah and Akai [48], observed a high incidence of water-borne diseases, such as diarrhea, cholera, and jaundice, amongst children and the elderly after flood events in Bangladesh and Ghana, respectively.

Furthermore, floods may destroy social capital in the form of endogenous social networks and contribute to the degradation of (fragile) ecosystems. When floods destroy social networks, it has been further observed that this too results in generalized destitution and a sense of grief among people who have lost loved ones, with psycho-traumatic consequences [40].

In summary, floods render agriculture-dependent households and (agrarian) communities more vulnerable to any adverse climatic and livelihood-depriving events, such as sickness and food consumption fluctuations; thereby, floods aggravate poverty and weaken resilience capacities [24,29].

Floods impact food insecurity [24,46] directly through the loss of household and farm assets, e.g., stored crops and livestock [8,35], but also indirectly through the loss of labor, either to death or forced migration, or to soil destruction and land degradation [24]; these may cumulatively culminate in a decline in food production [23,47]. Land contamination may impair food production further and, therefore, income, especially for agrarian households and communities who then witness short and long-term household food insecurity [24,48].

By way of summary, shocks such as floods often expose poor agrarian communities to negative livelihood effects and vulnerabilities. However, the answer to whether the effects correspond across space and time needs further research. We use several case studies to understand the effects of floods on the livelihoods of victims across space (three geo-ecological zones) and time (independent events).

## 3. Materials and Methods

### 3.1. The Study Sites

In total, five geo-ecological zones are found in Cameroon (see Figure 2): (1) The Sudano-Sahelian zone, (2) the Western Highlands (or Montane) zone, (3) the Humid Forest with monomodal rainfall; (4) The Humid Forest with bimodal rainfall, and (5) the High Guinean Savanna, in this study known as the Coastal zone [54].

In all of these geo-ecological zones, Cameroon has witnessed a significant increase in the frequency of floods over the last three decades [15,39]. Specifically, the frequency has increased from three per annum, in the 1980s, to a current average of five and up to ten in urban areas [55]. In 2020 alone, flooding impacted over 193,000 persons in Cameroon [40]. The most recent flood occurred in mid-August 2022 in the Far North Region, affecting approximately 40,000 people. The flooding was caused by heavy rainfalls, which caused rivers to overflow and dikes to break (see: floodlist.com/africa/cameroon-floods-farnorth-october-2022, accessed on 19 December 2022).

This study is concerned with three flood types that occurred in three separate geo-ecological zones: 'Complex' floods in the Sudano-Sahelian zone, coastal floods in the Coastal zone, and Riverine floods in the Western Highlands. Coastal floods result from strong winds or storms in coastal areas during high tides when low-lying areas are flooded by sea water. Riverine floods are characterized by gradual riverbank overflows that emanate from extensive rainfall over an extended period of time [53]. The area affected by river floods will depend on the size of the river and the amount of rainfall. Pluvial (or flash) floods occur in flat areas where the terrain cannot absorb the rain water, causing puddles and ponds. Though similar to urban flooding, pluvial floods occur mostly in rural areas, with serious impacts on the agricultural gainful activities and properties in the area [14,41]. We use the term 'complex' to describe a simultaneous occurrence of flood types. Flooding in the Far North Region of Cameroon (the Sudano-Sahelian zone) is complex, in that it is always a mix of the flash, riverine and pluvial types [14,37,41,56].

Empirical studies were carried out in the following three (of the five) geo-ecological zones that are highly exposed to flooding in Cameroon: the Sudano-Sahelian zone that covers the North and Far North Region (Zone I), the Coastal zone that covers the Littoral and South-West Regions (Zone II), and the Western Highlands (or Montane) zone (Zone III) in the North-West Region [37].

The Sudano-Sahelian zone corresponds to Zone I in Figure 2 and is characterized by a mean temperature of 28 °C and an average rainfall of 850 mm per annum; it has a base saturation of around 70%, and high amounts of weatherable soils. Settlement is concentrated along the rivers Benoue and Logone, which further exposes the population to regular floods [14,41]. In addition, the Ladgo dam, initially constructed to generate electricity, is now a regular source of flooding due to rapid sedimentation and improper management [41]. This zone is prone to both droughts and floods [14,41].

The Coastal zone corresponds to Zone IV in Figure 2, and comprises the Littoral and South-West Regions (with Douala and Buea as capitals, respectively), plus the Coastal edge of the South Region. It is the most important industrial and cash crop-producing zone in Cameroon. Due to this fact, it has the highest population density, of 66 persons per square kilometer and a high rate of vegetation loss [57]. The key rivers in this zone, notably Wouri, Dibamba, Mungo, Sanaga, Ntem, Manyu and Meme, often exceed their banks in the rainy season and generate floodplains, particularly along the multiple sandy beaches and cliffs [57,58]. The area experiences an average amount of rainfall, with 10,287 mm per annum, and exhibits warm temperatures year-round, ranging from 27 °C to 32 °C [14].

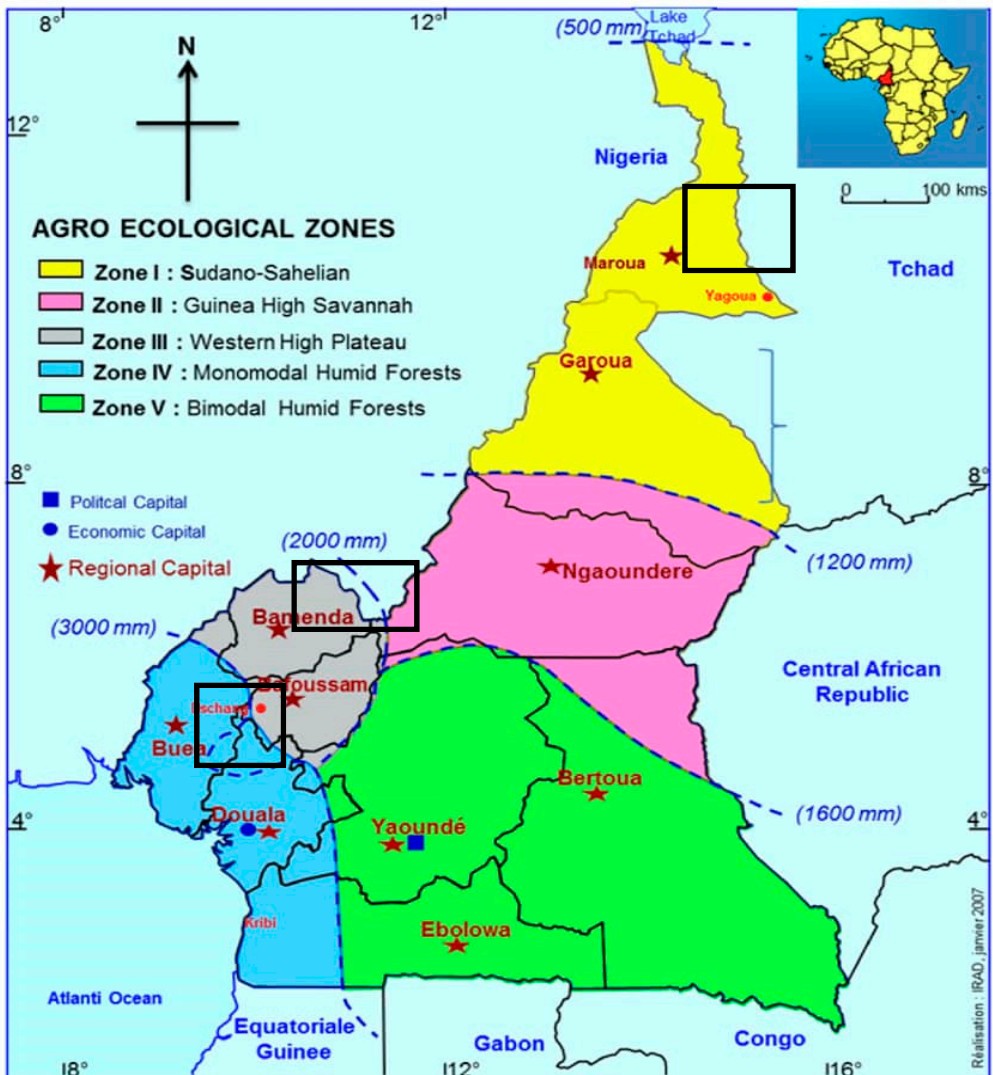

**Figure 2.** Geo-ecological zones in Cameroon and regions covered. Source: DSDSR ([59], p. 113). Notes: Black boxes represent the study regions in the three selected geo-ecological zones, represented by the regional capitals Maroua, Buea and Bamenda, respectively.

The Western Highlands, also known as the Montane zone, corresponds to Zone III in Figure 2. It covers the administrative districts of the West and North-West Regions, with the regional capital cities: Bamenda and Bafoussam. This area is of remarkable geological diversity and includes the Bamoun Plateau, which extends to an altitude of approximately 1240 m, the Bamiléké Plateau, which reaches an altitude of 2740 m through Mount Bamboutos, and the volcanic plateaus of Bamenda, at approximately 1800 m [57]. The landscape is characterized by medium mountains, savannah vegetation, stepped plateaus, low basins and plains, and patchy remnants of gallery forests, due to rapid deforestation in the last five decades [14,60]. Subsistence agriculture carried out on the mountain slopes creates favorable conditions for the rapid accumulation of large amounts of water, resulting in flooded areas especially during the rainy season. Zone III exhibits an average temperature of 21 °C and an average rainfall of 2500 mm per year [14]. Combined with the high altitude (1000–3011 m a.s.l.), this zone is frequently prone to flooding during the rainy season [14,41].

The selection of case study sites was accomplished In this manner for the following reasons: First, Cameroon is fondly called 'Africa in miniature', due to the presence of multiple representative geo-ecological zones. Second, it is also one of the regions most affected by natural disasters in SSA [38,40,61], thus lending itself as a good choice for

a comparative study. For instance, floods negatively affect, on average, 120,000 people every year, approximately 0.5% of the total population of Cameroon [62]. According to the CCKP [63], 65.4% (n = 17) of the registered major natural hazards in Cameroon are floods (1980–2020). This official estimate seems very conservative, since according to the data from the city administration, the economic capital Douala alone experienced over 300 floods between 1980 and 2014 [64]. Third, the three studied geo-ecological zones are those most affected by floods in Cameroon [40], and their economy is predominantly agriculture based [41]. Such a choice increases the potential of generated knowledge to inform practical flood policy decisions, with a focus on agriculture-dependent households. Lastly, each geo-ecological zone had recently experienced floods, for which our research team collected coherent impact assessment data, allowing for comparisons across the three geo-ecological zones [15,37,39] (see Appendix A for a schedule of floods and data collection periods).

The mixed-method design was adopted for all the original studies, during which quantitative and qualitative data were collected from flood victims in all three zones through face-to-face interviews. Overall, ten trained enumerators were involved in quantitative data collection using structured and pretested questionnaires. The questionnaires were tested on 5–10 respondents in the entry communities, to allow for adjustments before collecting data. A total of 2134 victims from 26 communities who were affected by five independent floods participated in the survey. These included 1000 flood victims from 11 communities in the Sudano-Sahelian zone (46.9%), 242 from 3 communities in the Coastal Zone (11.3%), and 892 from 12 communities in the Western Highlands (41.8%). The household heads responded to the questions. We assumed that they were well placed to recall the flood impacts. However, interviews took place at the homesteads, and other household members could participate.

*3.2. Sampling Approaches*

Only flood-affected households were interviewed to capture the direct flood effects. In each community, a sampling frame was obtained from the relevant local government authorities. From the list, flood-affected households were then identified, sampled, and interviewed.

Simple random sampling was applied to select the final sample only in the Far North Region (Sudano-Sahelian Zone I), where the number of victims was high; this was initially estimated at 20,000 [37]. A census approach was applied in all the flood communities, given the relatively small numbers of victims. Only victims who remained in the area and were willing to participate in the study were interviewed. The participation rate was between 90% and 100% for all flood victims, except in the Sudano-Sahelian zone, where our sample is estimated to be 5% of all those who were affected in the sampled villages by the selected floods.

In each community in the Sudano-Sahelian zone, the names/numbers of the affected households on the list were written on pieces of paper and randomly selected for interviews. Random selection, therefore, ended when the 1000th respondent was identified. A target of 1000 was set to optimize the shortcomings of time and logistics without losing quality. Household heads that were unavailable at the time of interview were replaced, still applying the randomization technique to the rest of the unselected households in the replacement list.

A mixed-method design was adopted for all the original studies, during which quantitative and qualitative data were collected from the flood victims in all three zones between 2012 and 2017. Overall, ten trained enumerators were involved in quantitative data collection using structured and pretested questionnaires. The questionnaires were tested on 5–10 respondents in the entry communities to allow for adjustments before collecting data. Data were collected after flood events, but not more than four months after the flood event. This upper limit was set to reduce difficulties with recollection, which comes with long waiting periods [14]. The interviews and data recording took place at the homesteads of the interviewees.

The key instrument used for data collection (structured questionnaire) comprised of four sections. Section one described the characteristics of the household head and his/her household; section two recorded the impact of the shock at the household level; section three captured the impact of floods at individual, household, and community levels; and section four identified the types of household response mechanisms. Each interview lasted between 10 and 15 min, depending on the level of damage incurred by the household. The structured questionnaires were complemented by field observations and key informant interviews. The collected data were entered, cleaned, and analyzed using SPSS (Statistical Package for Social Sciences), version 25.0. At a 95% confidence interval ($\alpha = 0.05$), both descriptive and statistical analyses are reported. The Chi-square distribution was the main statistical test used to compare the flood damage to households in the different zones.

ANOVA was also used to compare the age of the household heads, household sizes and the estimated monthly income of households in the different zones. As an analytical approach, ANOVA detects differences between (experimental) group means for selected variables (in this case, flood effects), with respect to one or more independent variables (geo-ecological zones); this was based on the assumptions that (1) the value of each observation is not influenced by that of other observations, and that (2) ANOVA is a good statistical application even for skewed data, as long as the sample size is large [65]. Sawyer [65] and Daniel [66] suggest a minimum group sample size of 30. As the size of each group meets this minimum criterion, ANOVA subdues any violations of homogeneity in variance assumptions.

The results were compared for households across geo-ecological zones. Given that these studies were carried out independently of each other, and that slight modifications were made to respond to contextual realities, only variables relevant to the respective capital assets captured in all the studies were compared. Therefore, for instance, financial capital was not analyzed. Where appropriate, the F-statistic was applied to explore the mean distribution of corresponding variables under the null hypothesis assumption, while the t-test was used to make statistical inferences on the differences of dependent variables between the geo-ecological zones [65]. A comparative analysis of the results between the three geo-ecological zones is presented and discussed in the next section.

### 3.3. Sample Size

A total of 2134 victims from 26 communities who were affected by five independent floods participated in the survey. These included 1000 flood victims from 11 communities in the Sudano-Sahelian zone (46.9%), 242 from 3 communities in the Coastal Zone (11.3%), and 892 from 12 communities in the Western Highlands (41.8%). The household heads responded to the questions. We assume that they are well placed to recall flood impacts. However, interviews took place at the homesteads, and other household members could participate. Table A1 in the Appendix A summarizes the database.

## 4. Results and Discussion

### 4.1. Descriptive Socio-Economic Household Results

The data indicated gender differences across the three geo-ecological zones. The majority of respondents (household heads) in both the Sudano-Sahelian zone (Zone I in Figure 2) and the Coastal zone (II) were male (close to 72% and 61%, respectively), while the majority of the respondents in the Western Highlands (Zone III) were female (over 53%). Although patriarchal systems exist in all zones, it seems to be stronger in the Moslem-dominated Zone I, compared to the Christian-dominated Zone II. These gender-based differences can have implications on flood perception, resilience, and the eventual impacts on livelihoods, given that women generally have limited access to, and control over, the resources needed to prepare for, or respond to extreme events [8,67].

With respect to educational achievement, over 38% of the entire sample had finished primary school (64% in Zone I, 38% in II, as well as III). Approximately 17% of the respondents in Zone I, 33% in Zone II, and 44% in Zone III had completed secondary school

education. Buchenrieder et al. [8] and Shah et al. [56] report that differences in educational levels, for instance, will have an effect on the management approach to floods and their effects on livelihoods. Previous studies indicate that persons or communities with low educational levels tend to adopt more low-tech and ad hoc flood management strategies, such as temporary migration, gift economy, and mutual aid [8]; meanwhile, more educated ones are likely to adopt high-tech solutions, such as early warning systems and the construction of retention dikes [5,15,54].

The main sectors of employment are summarized in Figure 3. It reveals differences across zones. While only a slight difference was observed for the Sudano-Sahelian zone (Zone I: 49% employed in the farm sector and 51% in the non-farm sector), the gap was greater in the Coastal zone (Zone II: 20% employed in the farm sector and 80% in the non-farm sector) and the Western Highlands (Zone III: 59% employed in the farm and 41% in the non-farm sector. It is plausible to assume that the border with Nigeria in Zone I and the economic capital of Douala, which hosts a seaport nearby Zone 2, encourage non-farm activities. In addition, the damage to the agricultural income base, due to the increasingly experienced floods, might create disincentives to engage in agriculture [2,8,19,41]. Zone III is land-locked and driven by an agrarian economy, with fewer non-farm opportunities [15]. All households, irrespective of the main income-creating occupation, depend at least partly on agriculture for their livelihoods. Any flood effects on the agricultural sector are, therefore, likely to influence livelihood outcomes in the studied communities, given their reliance on agriculture for subsistence and cash income [14,41].

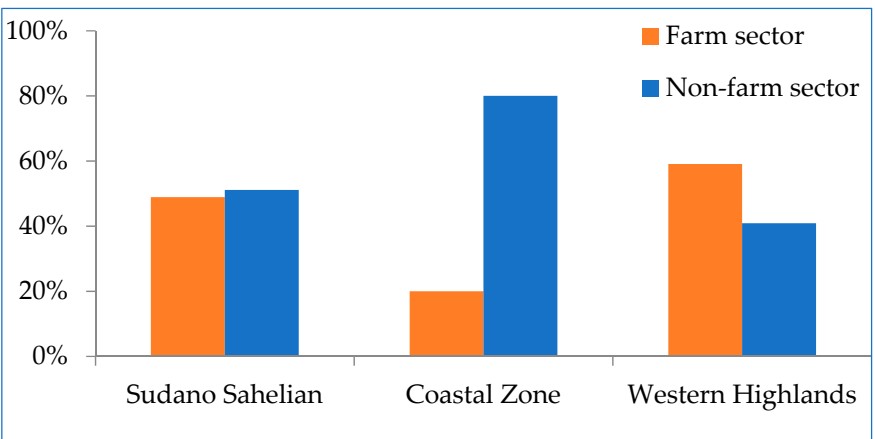

**Figure 3.** Main sector of employment of household head, per geo-ecological zone.

Analyses of additional socio-demographic variables are presented in Table 1. The mean age of the respondents was significantly higher in the Coastal zone (Zone II), compared to the Western Highlands (Zone III) and the Sudano-Sahelian zone (Zone I): 45 years, 38 years, and 33 years, respectively. The mean age for the entire sample (39 years) is not very different from the mean of 41 years reported in a recent study on Cameroon [8]. Differences were further observed in mean household sizes, with Zone III recording the largest household size, followed by Zone I and then Zone II (8, 7, and 6 household members, respectively). The mean monthly household income in Zone III was significantly higher compared to Zone I and Zone II (FCFA183,710 [US$ 288], compared to FCFA55,780 [US$ 87.5], and FCFA89,300 [US$ 140.1], respectively). It is, therefore, expected that, given the older age of the household heads and the higher household incomes, the flood victims in the Western Highlands (Zone III) will better cope with floods, compared to those in the other geo-ecological zones.

**Table 1.** Demographic characteristics of households by geo-ecological zone.

| Variables | Geo-Ecological Zone | Mean | Std. Deviation | ANOVA |
|---|---|---|---|---|
| Age of household head (in years) | Sudano Sahelian (I) | 32.72 | 11.067 | F = 99.783 |
| | Coastal (II) | 45.47 | 12.096 | $p = 0.000$ |
| | Western Highlands (III) | 37.68 | 14.172 | |
| Household size (in persons) | Sudano Sahelian (I) | 7 | 6 | F = 24.729 |
| | Coastal (II) | 6 | 3 | $p = 0.000$ |
| | Western Highlands (III) | 8 | 5 | |
| Estimated monthly Income (in FCFA) | Sudano Sahelian (I) | 55,780 | 28,810 | F = 67.837 |
| | Coastal (II) | 89,300 | 88,470 | $p = 0.000$ |
| | Western Highlands (III) | 183,710 | 351,430 | |

Note: 1 US$ = 637.25 FCFA (www.xe.com, 14 August 2022).

*4.2. Impact of Floods on Household Livelihoods*

Subjective responses from respondents revealed that most of them perceived the impacts of the different floods as highly negative, irrespective of the year and place of occurrence (74% in the Sudano-Sahelian, 49% in the Coastal, and 86% in the Western Highlands zone) (Figure 4). To enhance the consistency in quantifying losses for the multiple case studies in different geo-ecological zones and largely in line with the capital portfolio of the SLF [22], we limit the analysis to (1) loss of productive capital, (2) effects on human capital, and (3) damage to private and public property, such as houses and infrastructure.

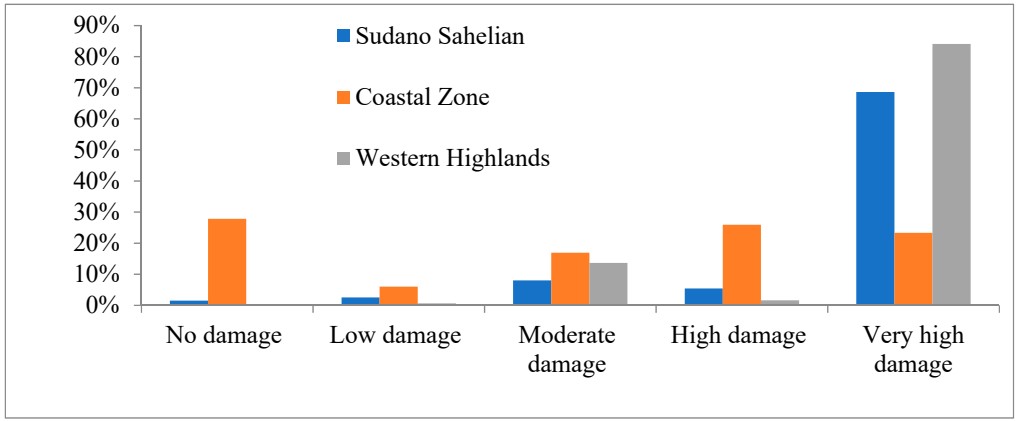

**Figure 4.** Level of flood damage to households, percentage of household heads.

4.2.1. Loss of Productive Assets

Obviously, agriculture-dependent households depend on arable land for production. The destruction of farmland by flooding was thus considered a major disaster. Erosion of the top soil due to flooding occurred in all zones, which rendered land less fit for farming activities. However, this was more intense in Zones I and II than in Zone III.

Other productive assets, whose losses were captured in all case studies related to the loss of livestock and damage to crops (Figure 5), were noted. The livestock losses after flooding were higher in the Western Highlands (by 98% of respondents) and the Sudano-Sahelian zone (by over 95% of respondents), as opposed to only 47% of the household respondents in the Coastal zone. The scenario was different with regard to crop damages. Significantly higher crop damages were recorded in the Sudano-Sahelian zone (reported by over 88%), the Coastal zone (over 53%) and lastly the Western Highlands (34% of the households). Overall, the farm sector (both crop and livestock) was seriously affected by the floods across all geo-ecological zones, even if the effects were significantly higher in the Sudano-Sahelian zone.

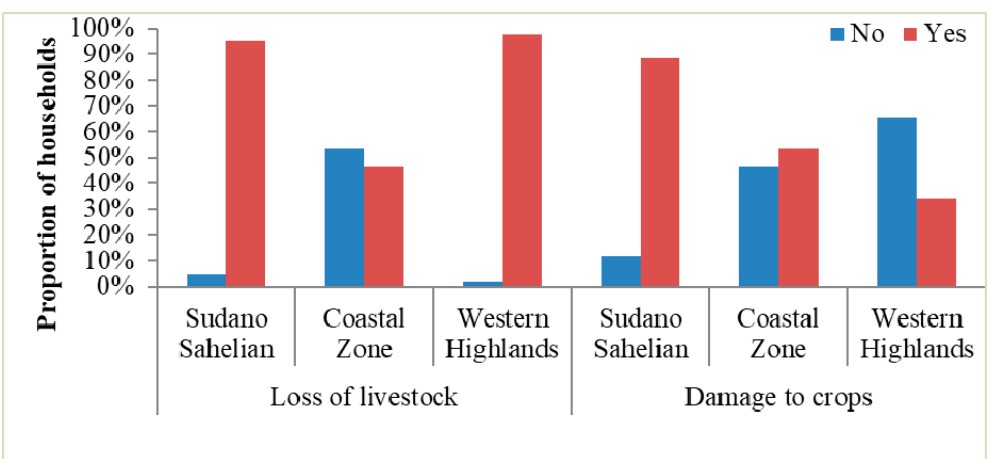

**Figure 5.** Loss of and damage to household productive assets, share of households. Notes: Chi-square livestock = 622.77, *p* = 0.000; Chi-square crop = 537.676, *p* = 0.000.

In total, 72% of all respondents reported generally impaired economic activities due to the experienced floods. The share of reports, however, varied in Zone III, Zone II, and I (85%, 69% and 62%, respectively).

### 4.2.2. Flood Effects on Human Capital

Human capital was captured in the 26 case studies, through health challenges, injury, and death, attributable to floods (Table 2). Empirical results suggest that less than 33% of all the respondents experienced deteriorating health conditions after floods, irrespective of the geo-ecological zone. The most common health deficit observed was diarrhea, emanating from contaminated drinking water sources. This effect was reported more often in the Coastal zone (57.5%), compared to the Sudano-Sahelian and Western Highlands zones (29% and 12%, respectively). Consistent with the previous results (e.g., [27,54]), a higher share of respondents in Zone I reported physical injuries from floods, compared to Zones II and III (28% and 19.5%, respectively). However, the death of household members as a result of the flooding was higher in Zone III than in Zones I and II (74%, 49% and 13%, respectively). Although the cumulative human capital effects are high for all the geo-ecological zones, the specific effects on human capital seem to vary. This may be attributed to varying levels of preparedness and disaster management capacity; nevertheless, this would have to be investigated further.

**Table 2.** Flood effect on human capital.

| Human Capital Variables | Geo-Ecological Zone | In % | | Chi-Sqaure |
|---|---|---|---|---|
| | | **No** | **Yes** | |
| Increase in sickness | Sudano Sahelian (I) | 70.9 | 29.1 | $X^2 = 211.054$ |
| | Coastal (II) | 42.5 | 57.5 | $p = 0.019$ |
| | Western Highlands (III) | 88.2 | 11.8 | |
| Physical injury | Sudano Sahelian (I) | 72.0 | 28.0 | $X^2 = 65.79$ |
| | Coastal (II) | 80.5 | 19.5 | $p = 0.000$ |
| | Western Highlands (III) | 88.3 | 11.7 | |
| Loss of life from direct flooding | Sudano Sahelian (I) | 51.4 | 48.6 | $X^2 = 298.105$ |
| | Coastal (II) | 86.8 | 13.2 | $p = 0.000$ |
| | Western Highlands (III) | 26.2 | 73.8 | |
| Loss of economic activities | Sudano Sahelian (I) | 37.9 | 62.1 | $X^2 = 108.685$ |
| | Coastal (II) | 30.8 | 69.2 | $p = 0.000$ |
| | Western Highlands (III) | 14.8 | 85.2 | |

### 4.2.3. Damage to Property

Figure 6 presents the summary of damages to private and public property across all the case studies and by geo-ecological zones. A significantly higher proportion of respondents in the Sudano-Sahelian zone stated to have not experienced property damages after floods, compared to their counterparts in the Coastal and the Western Highlands zones (94%, 23%, and 86%, respectively). Observations during data collection revealed that most structures in Zones I and II, particularly housing, were of rather poor quality, rendering them more vulnerable and susceptible to flood damages than households in Zone III. Bang et al. [41] attribute high property damage in Zone I to the highly weatherable soils that are particularly vulnerable to frequent floods. The Western Highlands (Zone III) suffered more from damage to public infrastructure, particularly roads, when compared to Zones II and I (83%, 50% and 59%, respectively). Field observations confirmed this trend in property damages in the three geo-ecological zones.

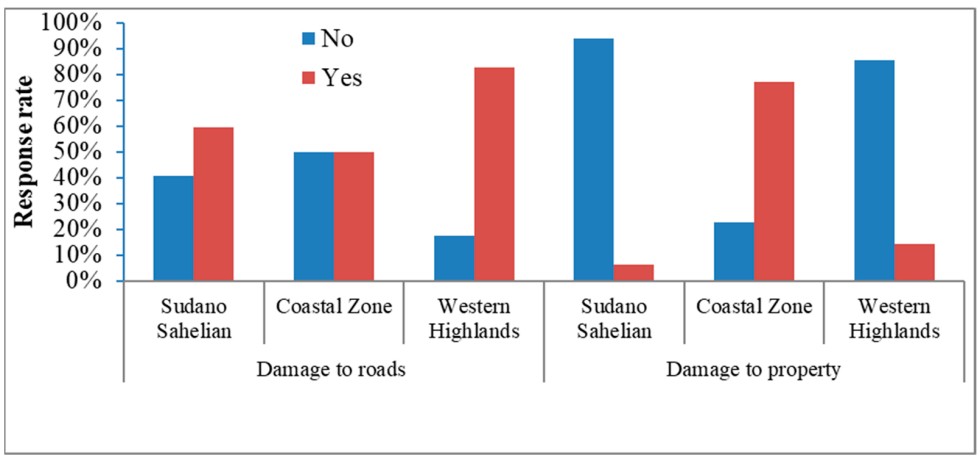

**Figure 6.** Damage to infrastructure. Notes: Chi-square livestock = 622.77, *p* = 0.000; Chi-square crop = 537.676, *p* = 0.000.

### 4.3. Discussion

The growing research on floods in SSA has greatly neglected the modeling of multiple floods and their impacts on livelihoods across space and time. We have contributed to this vacuum by analyzing the effects of five independent case study floods that affected 26 communities located in three geo-ecological zones in Cameroon; these floods occurred between 2012 and 2017.

The majority of the respondents (household heads) in two of the three geo-ecological zones were men. Their share was significantly higher for the Sudano-Sahelian zone (72%) and the Coastal zone (62%, *p* = 0.000). While male household dominance is sustained by a strong patriarchal system in Cameroon [39], the fact that household heads are more often male in the Sudano-Sahelian Zone might be linked to gender roles based on Islam [68]. Nevertheless, it also could simply be a result of men being more present at the homestead in this zone due to their on-farm income earning activities. Including gender differences is, therefore, central for effective flood preparedness and management in all of the studied geo-ecological zones. A reasonable proportion of the respondents (victims) had only primary education (almost two thirds and one-third in Zone I and II and III, respectively). Both statistics fall below the national literacy rate of 77.1%, signaling that those hit by extreme events, such as floods, have most often only finished primary school (see also [38]). School education may influence the perception of the victims vis-à-vis the offered risk management schemes and the willingness to adopt them [8]. However, differences across zones suggest different pathways towards flood risk preparedness and management, as high-tech solutions are likely to be less feasible for Zone I, compared to Zone II and III. Over 60% of all the respondents were married with family. This result is in favor of also

propagating informal, endogenous, community-based approaches to flood management in SSA (as pointed out in [37,69]). Such an approach can compensate, to some degree, for the formal institutional deficits identified.

The agricultural sector was highly affected by floods, irrespective of the community or geo-ecological zone. The loss of livestock (95%, 47%, and 98%) and on-farm crop damages (88%, 53%, and 34%) were reported as a result of flooding in Zones I, II and III, respectively. Usually, stored seeds are also destroyed. However, the effects were consistently and significantly higher in the Sudano-Sahelian zone (I), compared to the Coastal and the Western Highlands zones (II and III). These findings relate with the works of [2,15,19,47,50], who contend that floods cause severe problems in all farm regions. Ainuddin et al. [4], for instance, report a 98.3% loss of crops in the North Western Zone of Ethiopia following flood events. Zhang et al. [70] report similar trends across China, with no firm spatiotemporal pattern of flood-destroyed crops across regions. Furthermore, flood effects on livestock and crops translate into subsistence and monetary income losses. This directly and indirectly decreases access to food, resulting in short- and long-term food insecurity in all the geo-ecological zones (see also [3,46,48,49]).

Flood effects on human capital are high across the board, but the specific type seems to vary notably among the three geo-ecological zones. In terms of human capital losses, approximately one-third of the sample reported an increase in health hazards after flooding. The most common health issue was diarrhea, which was reported by almost 56% of the respondents in the Coastal zone. Rising health hazards from flooding have also been reported in other recent studies in SSA (e.g., [1,27,54]). For instance, Suhr and Steinert [27] find that the majority of studies point to an increased risk of infection with cholera, scabies, and other diseases from floods, based on their systematic review of 2603 studies on the epidemiology of floods in SSA.

A significantly higher proportion of the household members in the Sudano-Sahelian zone (28%) reported physical injuries from floods compared to the Coastal and the Western Highlands zones, probably due to the poor quality of housing and road infrastructure, whose damage might have inflicted injuries on the victims [34,40] as they easily succumb to floods. This is plausible, given that the Sudano-Sahelian zone hosts the two poorest regions in Cameroon [41,53]. Njogu [53] contends that flood effects on infrastructure in this zone are a combined effect of high poverty rates, and social, biophysical and place vulnerability.

Many scholars (e.g., [8,11,17–19,30,55]) have also revealed that the death of victims is the most devastating consequence of natural disasters. More deaths from flooding were recorded in the Western Highlands (Zone III) than Zones I and II (see Table 2). This is probably due to a lack of experience with floods, which can lead to inadequate preparedness. Nevertheless, this finding corroborates the existing evidence that flash floods account for the highest average mortality among all flood types, even if spatial variations exist [71]. It is likely that experiential knowledge is at work, especially in Zones I and III, where flood frequency is higher [40] and community-based strategies have been developed to cope with the frequent occurrence of floods [1,8,37,41].

A greater proportion of all households reported impaired economic activities. However, economic loss was significantly higher for the households in the Western Highlands (III) than the Coastal (II) and the Sudano-Sahelian (I) zones (85%, 69% and 62%, respectively). We might see here a combined effect of inexperience, inadequate flood preparedness, and the possible damage of rapid, onset floods on agriculture, on which a majority of livelihoods in Zone III depend [8,14,23]. Economic losses from floods were reported in a study of six regional floods in Bangladesh between 2004 and 2007, specifically in the form of income and employment losses from the agricultural sector [72]. Differences in economic losses across case studies reiterate the importance of contextualizing variables for capturing the economic effects of floods. However, all studies report negative outcomes, irrespective of the measurement variables applied.

A significantly higher proportion of households in the Sudano-Sahelian zone reported damages to physical assets after floods, compared to the Coastal and the Western Highlands

zones (94%, 23%, and 86%, respectively). Due to the hilly nature of the terrain in the Western Highlands (Zone III), the damage to roads was more severe when compared to Zone II and I (83%, 50%, and 59%, respectively). The study of Okunola [73] in South Africa also recorded significant damages to homesteads and public infrastructure, as a result of the April 2022 floods that swept across Kwa Zulu-Natal Province. Over 2000 houses and 4000 'informal' homes or shacks were severely damaged. Roads, more than 200 schools, and communication, water, and electricity systems were impaired. Similar results have been reported for Malawi [53], for dairy farm infrastructure in New Zealand [74], and for agricultural land damage after floods in Pakistan [75]. Despite differences in the physical farm assets, they were impaired by floods, with the effects seemingly related to the resilience of the critical infrastructure prior to the floods.

## 5. Conclusions

Very often, floods inflict negative effects on livelihoods, infrastructure, and ecosystems. Flood risk management is particularly important in SSA where (1) poverty is endemic, (2) early warning systems usually do not exist, and (3) the capacity of disaster management institutions is weak, dysfunctional, or simply absent. In this context, this article contributes to the growing literature on floods and livelihoods in SSA, by drawing on a sample of 2134 victims of independent floods in 26 communities; this was in order to examine the effects of multiple floods across three geo-ecological zones in Cameroon on agriculture-dependent livelihoods. The results led to a number of conclusions.

First, agriculture-dependent livelihoods were negatively affected across all the geo-ecological zones, albeit at different magnitudes. This seems normal, given that agriculture is the major source of livelihoods in the three studied geo-ecological conditions. Second, apart from economic losses, which were high in all the geo-ecological zones, a high regional variance was observed for other parameters, such health hazards, physical injuries, and the loss of human lives. This seems to suggest that the contextual (specific geo-ecologic) approach to understanding floods and their impacts should be further strengthened. In our study, the negative effects were more pronounced in the Sudano-Sahelian zone, with the lowest average annual rainfall; this is compared to the Coastal and Western Highlands zones, with a higher average rainfall. We assume that this is due to the high soil saturation and weatherable soil quality, and the less resilient construction of the houses. Third, flood-related deaths were significantly higher in the Western Highlands, compared to the Coastal and the Sudano-Sahelian zones; this is probably due to a lower flood risk perception, which may have caused inadequate preparedness.

This study, therefore, demonstrates how analyzing flood effects across space and time can provide insights into strategies that enhance broad-based efforts towards preventing, mitigating, managing, and developing private and public sector resilience against floods. The findings suggest that national and international policies (e.g., the Sendai Framework for Disaster Risk Reduction) need to be flexible enough to introduce contextual specificities into successful flood management. In other words, establishing broad-based flood policies can benefit from local realities, which differ across geo-ecological zones. However, as observed in this study, some aspects (agricultural damages) can be consistently similar across geographic space and time. Forging ahead with such a research agenda could generate vital insights into the area of study, which can support SSA countries and the subcontinent to develop and implement successful flood management policies; these are urgently required to deal with the surging number of floods and their effects.

Expanding this research agenda to other extreme events, such as droughts, whose frequency has also increased over the past 50 years [76], can support the continent in developing informed disaster preparedness and management policies; these are needed to shape disaster management in a continent characterized by a weak state and with a market capacity for disaster risk reduction. For such a research agenda to have an optimal impact, the need to harmonize data collection instruments and methods across space and time for effective comparisons cannot be overemphasized. This is a prerequisite for making policy

suggestions to support SSA's flood risk reduction capacity, consolidating its achievements towards the globally designed Sustainable Development Goals, while reducing the harm caused to agriculture-dependent households.

**Author Contributions:** R.A.B. coordinated the work, from field research to the paper write up. He also interpreted the results and made conclusive suggestions. K.A.N. participated in the field data collection and ran the analysis. G.R.B. and J.N.K. reviewed the literature and critically commented on the multiple drafts. All authors contributed to critically fine-tuning the paper. All authors have read and agreed to the published version of the manuscript.

**Funding:** This research was funded under the post-doctoral fellowship program (2013–2018) by the Volkswagen Foundation, Germany, grant numbers 89866 & 91085.

**Data Availability Statement:** The data that support the findings of this study are available from R.A. Balgah upon reasonable request.

**Acknowledgments:** The authors acknowledge the financial support of the Volkswagen Foundation Germany, for the field work summarized in this article. We are also grateful to the flood victims and enumerators for providing and helping in data collection, respectively. Special thanks go to the anonymous referees for improving the quality of the final paper. We thank the three anonymous reviewers for their careful reading of our manuscript and their many insightful comments and suggestions. As always, any remaining errors or oversights are mine alone.

**Conflicts of Interest:** The authors declare no conflict of interest.

## Appendix A

**Table A1.** Schedule of floods and data collection periods.

| Geo-Ecological Zone | Community | Respondents | Flood Date | Data Collection Date |
|---|---|---|---|---|
| Sudano-Sahelian zone (Zone I) | Bazala | 148 | August 2016 | November 2016–January 2017 |
| | Gazawa | 99 | | |
| | Kaikai | 82 | | |
| | Gobo | 74 | | |
| | Katoual | 56 | | |
| | Maga | 113 | | |
| | Mora | 86 | | |
| | Pouss | 107 | | |
| | Ziling | 71 | | |
| | Zoubouk | 78 | | |
| | Zongoya | 86 | | |
| | **Total** | **1000** | | |
| Coastal zone (Zone II) | Bekora | 68 | September 2016 | December 2016 |
| | Clerks quarters | 69 | | |
| | Motowoh | 105 | | |
| | **Total** | **242** | | |
| Western Highlands zone (Zone III) | Baba I | 71 | September 2015 | October 2015 |
| | Babessi | 77 | September 2012 | December 2012 |
| | Gayama | 92 | | |
| | Kpep | 70 | | |
| | Akum | 105 | | |
| | Ambo | 62 | | |
| | Bado | 60 | | |
| | Edzong | 67 | August 2017 | October–November 2017 |
| | Ifung | 70 | | |
| | Munka | 84 | | |
| | Munkep | 60 | | |
| | Ogim | 74 | | |
| | **Total** | **892** | | |

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
