# Peer review of "Impacts of Floods on Agriculture-Dependent Livelihoods in Sub-Saharan Africa: An Assessment from Multiple Geo-Ecological Zones"

_land, doi:10.3390/land12020334_

Round 1
Reviewer 1 Report
This manuscript attempts a comparative study of the effects of flooding in different geo-ecological zones in Cameroon as a means of using this type of analysis to help develop policies and practices to minimize the disruption caused by flooding. The research design is innovative with an attempt to use similar methods across a long time frame to assess local responses to flooding in different regions of the country. Issues to arrive in the presentation and analysis though. First, ti would help to provide more background on both the regions included in the analysis and 5 individual events here. More detail about the regions and areas would allow for more nuanced judgements of the effects of the disasters. In the few instances where specific details are used, I would suggest the authors rethink their assumptions. I would suggest socio-economic factors are probably more important than religion is explaining why more female headed households were interviewed in the coastal and highlands regions than in the Savanna region. Finally some background on the specific events analyzed would allow for some gauge of one factor the authors do not mention, the relative severity of the "shock." I do think this research is publishable, but I do believe the authors need to think closely about the factors need to complete their intended analysis.
Author Response
We thank the reviewers for their comments and suggestions regarding our paper. They have further motivated us to rethink and clearly articulate the contribution of our paper. The revisions to our paper have been done accordingly.
The revisions are highlighted in the manuscript. Minor revisions, e.g. typos are not highlighted to ease reading.
Three of the four authors are native English speakers. Nevertheless, an English language check was undertaken for the article as suggested by the reviewers. Furthermore, newly introduced literature is marked in yellow in the text and in the reference list.
In the following, we explain how we have addressed the concerns raised and incorporated the suggestions into the revision. We cite where we effected these changes in the updated manuscript.

Reviewer 2 Report
The history of mankind is a history of fighting against disasters. Based on primary survey data, the authors focus on the impact of flooding on agriculture-related livelihoods. In general, the study has certain significance, but there are still some deficiencies. Here are some suggestions for your reference:
(1)The introduction needs a modest rewrite. First, the author needs to make a systematic definition of the core concept, especially the concept of agriculture-related livelihood. If this definition is not clear, the subsequent theoretical analysis and result analysis will be unclear. Second, compared with the existing research, the marginal contribution of this research is not clear. Third, the key scientific questions to be addressed by this study are also unclear.
(2) The research lacks in-depth theoretical analysis. On the one hand, relevant literature review is not in place. On the other hand, there is a lack of dialogue with classical theories and studies.
(3) Because the core concepts are not clearly defined, part of the overall result analysis is not very meaningful. The author simply made a few comparison graphs and could not get the effect of the flood on livelihoods shown in the title.
(4) The discussion section also needs to be further developed, and there is a lack of comparison with similar research results.
Author Response

(The authors gave the same response as above.)

Reviewer 3 Report
Impacts of Floods on Agriculture-Dependent Livelihoods in Sub-Saharan Africa. An Assessment from multiple Geo-Ecological Zones
Abstract
1- What do the authors mean in the sentence: Unfortunately, this “scholarship” is characterized..., in this case it would be: Unfortunately, this “research” is characterized ...? – In the same sentence, do the authors inform that the research cases are “isolated” and of “little relevance”? You see, just by reading this sentence, which needs to be better written, I could make the decision to recommend rejection of the manuscript. It is noticed that much of the manuscript needs to be better evaluated and rewritten.
2- How can there be a single sample of 2,134 flood victims from 26 communities? How can a single sample represent the complexity of the interaction and effects of floods on the means of production?
3- The rest of the summary needs to be equally improved in order to better highlight the main results found and possible recommendations.
1. Introduction
L 40-41 – I did not understand this part: “ex-post or ex-ante” - needs to be better evaluated and rewritten.
- Why use the word “schock” if you can refer to floods only as an “extreme event”?
- The last paragraph needs to be modified, listing the topics of each piece of information: “The section 2 presents a concise overview …”
3.1. The study sites
- We need a separate figure in (a) that shows us the five regions of Cameroon: (1) The Sudano-Sahelian Zone, (2) the Western Highlands (or Montane zone), (3) The Humid Forest with Monomodal Rainfall; (4) The Humid Forest with Bimodal Rainfall, and (5) The High Guinean Savanna; and (b) that shows us the three geoecological zones: (1) Complex floods in the Sahel Zone, (2) Coastal floods in the Coastal zone, and (3) Riverine floods in the Western Highlands.
- A quote from a regional newspaper is not enough to attest to the claims of an increase in the number of floods since the 1980s. Besides, the link no longer refers to any information (www.africanews.com/2022/10/13 /cameroons-far-north-region-faces-devastating-floods/). Is this data not recorded by any government agency, for example? Where does the information on the increase in trends in the number of floods and the number of people affected by them come from officially?
- Where does the average information for the variables mentioned in the fourth paragraph of this section come from? (Measures of relative humidity, temperature and annual precipitation)? Is there any official body of the Government of Cameroon that makes this data/information available? Are they based on station measurements with which spatial distribution in each zone? What is the quality of this data in terms of continuity, percentage of failures and homogeneity?
- The information on the characteristics of the types of floods in each geoecological zone, in the second paragraph, needs to be accompanied by information on the relief, so a figure that characterizes the geomorphology of these three zones is essential.
3.2. Sampling Approaches
- It is necessary to improve the description of the statistical methodology used to extract sample results, with emphasis on the ANOVA F test.
Final consideration:
Estimates of the damage caused by floods to the most vulnerable population in the three areas chosen for the study have an important social value for the implementation of public policies. I suggest that the authors highlight in the manuscript, in the final sections, which strategy they intend to address so that this type of study really reaches decision makers, who can truly make use of them to implement systems to mitigate and reduce damage to such populations.
Author Response

(The authors gave the same response as above.)

Round 2
Reviewer 2 Report
I have no other comments, thank you.
Author Response
Thank you for your favourable response.
Reviewer 3 Report
I appreciate the attention with which each question I mentioned was answered. The article got better with it.
My only observation and final recommendation is to better separate what is discussion from what is conclusion. In conclusions we should not, as a rule, cite references, and this happens, which makes me wonder whether these sentences with references should not be moved to the previous session of discussions. It's just a suggestion.
Author Response
We thank the reviewers for their favorable comments and final suggestions regarding our paper.
The revisions are highlighted in the R2-manuscript. Minor revisions, e.g. typos are not highlighted to ease reading.
In the following, we explain how we have addressed the suggestions of Reviewer 3.
"I appreciate the attention with which each question I mentioned was answered. The article got better with it."
Thank you very much for this comment.
---
"My only observation and final recommendation is to better separate what is discussion from what is conclusion. "
We have deleted sections from the discussion that might have given the impression to be conclusions. Furthermore, we have rewritten parts f the discussion and conclusion to address your critique.
---
"In conclusions we should not, as a rule, cite references, and this happens, which makes me wonder whether these sentences with references should not be moved to the previous session of discussions. It's just a suggestion."
We have reconsidered whether the mentioned references are truly necessary in the conclusion and have removed all except one. Thank you for reminding us of this convention.
